# Theoretical investigation of active listening behavior based on the echolocation of CF-FM bats

**Takahiro Hiraga**[1], **Yasufumi Yamada**[2]*, **Ryo Kobayashi**[2]

**1** Department of Mathematical and Life Sciences, Hiroshima University, Department of Sciences, Higashi-Hiroshima, Japan, **2** Program of Mathematical and Life Sciences, Hiroshima University, Department of Sciences, Higashi-Hiroshima, Japan

* yasufumi.yamada1@gmail.com

## Abstract

Bats perceive the three-dimensional environment by emitting ultrasound pulses from their nose or mouth and receiving echoes through both ears. To determine the position of a target object, it is necessary to know the distance and direction of the target. Certain bat species that use a combined signal of long constant frequency and short frequency modulated ultrasounds synchronize their pinnae movement with pulse emission, and this behavior has been regarded as helpful for localizing the elevation angle of a reflective sound source. However, the significance of bats' ear motions remains unclear. In this study, we construct a model of an active listening system including the motion of the ears, and conduct mathematical investigations to clarify the importance of ear motion in direction detection of the reflective sound source. In the simulations, direction detection under rigid ear movements with interaural level differences was mathematically investigated by assuming that bats accomplish direction detection using the amplitude modulation in the echoes caused by ear movements. In particular, the ear motion conditions required for direction detection are theoretically investigated through exhaustive simulations of the pseudo-motion of the ears, rather than simulations of the actual ear motions of bats. The theory suggests that only certain ear motions, namely three-axis rotation, allow for accurate and robust direction detection. Our theoretical analysis also strongly supports the behavior whereby bats move their pinnae in the antiphase mode. In addition, we suggest that simple shaped hearing directionality and well-selected uncomplicated ear motions are sufficient to achieve precise and robust direction detection. Our findings and mathematical approach have the potential to be used in the design of active sensing systems in various engineering fields.

## Author summary

Many mammals use visual sensing for the primary perception of their surroundings, whereas bats accomplish spatial perception by active acoustic sensing. In particular, by emitting ultrasound pulses and listening to the echoes, bats localize reflective objects, a process known as echolocation. Certain bat species move both of their ears while receiving

**Funding:** This work was supported by the Japan Society for the Promotion of Science KAKENHI Grant Numbers 19K15012, 22K14278 (Grant-in-Aid for Early-Career Scientists to YY), and 21H05170 (Grant-in-Aid for Transformative Research Areas(B) to YY). No author has received a salary from our funders. The funders had no role in study design, data collection and analysis, decision to publish, or preparation of the manuscript.

**Competing interests:** The authors have declared that no competing interests exist.

the echoes, but the precise type of ear movements that facilitate direction detection remains unclear. In particular, although ear movements have been investigated through practical demonstration or simulations based on actual behavioral measurements, there has been little theoretical consideration of the kinematic requirements for amplifying spatial information based on an objective viewpoint.

This paper describes a simple mathematical model for investigating the active listening strategy employed by bats. The theory suggests that certain ear motions might enable highly accurate direction detection that is robust to observation errors. In addition, we determine what kind of ear motions are optimal for direction detection of reflective sound sources. This study not only theoretically supports the significance of pinnae motion in bats, but also opens up the possibility of engineering applications for active listening systems.

## Introduction

Echolocation describes an active acoustic sensing capability whereby the surroundings can be imaged using the echoes from sound emissions. Bats perceive the three-dimensional (3D) environment through echolocation with high-frequency ultrasound [1]. Despite the simple sensing design, i.e., only one transmitter (mouth or nose) and two receivers (left and right ears), bats accomplish precise navigation tasks such as the pursuit of prey [2,3] and flying together with multiple conspecifics [4,5]. The highly sophisticated mechanisms that enable 3D navigation with ultrasound have attracted extensive and longstanding attention from physiological and behavioral scientists.

To date, the acoustic imaging process in the auditory system of bats and other animals has been widely investigated [6–8]. Previous studies have reported that bats have an encoding mechanism for the interaural sound pressure level difference (ILD) in the lateral superior olive, as seen in many mammals [8–13]. The lateral superior olive in bats is larger than in other mammals [14], and acoustic localization with ILD is physically suited to less-diffractive high-frequency sound. Thus, the ILD encoding mechanism is regarded as one of the key properties whereby bats detect the echo source direction. Recent studies have conducted more comprehensive analysis combining ILD mechanisms with head-related transfer functions [15–17]. These functions are important features that describe the echo strength as a function of the echo source direction. Measurements of head-related transfer functions in various bat species suggest that the pinnae are used for beamforming of the echoes reflected from objects [15–18].

This is not the only key evolutionary feature that bats have acquired for acoustic localization. Several species of bats employ behavioral solutions for echo source direction detection. The *Rhinolophidae* and *Hipposideridae* families synchronize the movement of their left and right pinnae with the pulse emissions [19–22]. This active listening behavior has been reported for constant frequency–frequency modulated (CF-FM) bats, who use a compound signal consisting of a CF part and an FM part (Fig 1A). Previous physiological and ethological studies have clarified that CF-FM bats detect the precise time interval between pulse emission and echo arrival using the FM part, allowing them to measure the distance to the object accurately [7,23]. The CF part is used for fluttering moth detection and Doppler shift compensation [24–26]. According to measurements from *Rhinolophus ferrumequinum*, both pinnae move continuously while listening to the CF part of the echo [20]. Chamber experiments investigating the obstacle avoidance flight of bats have shown that elevation angle detection performance is significantly decreased when bats are prevented from moving their pinnae [27]. Elevation angle

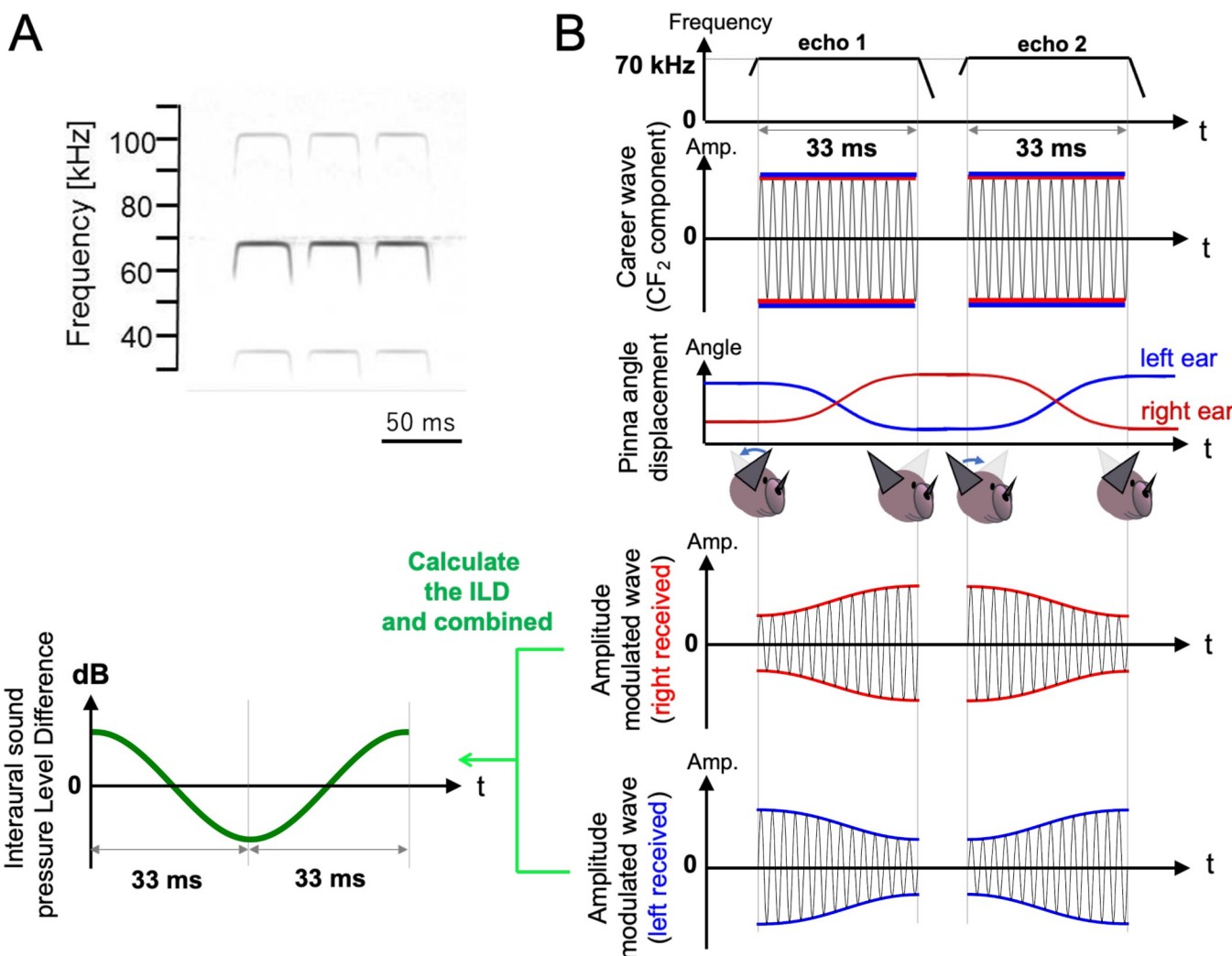

**Fig 1. Pulse emission and reflective echo patterns of the simulation.** (A) Typical time–frequency structure of the echolocation pulse emitted by *Rhinolophus ferrumequinum Nippon*. (B) Schematic diagram of amplitude modulation of $CF_2$ component in the simulated echo caused by virtual pinnae motions. In this example simulation, the left and right ear movements were assumed to exhibit anti-phase motions in pitch angle over ±15˚. The hearing axis in the right ear rotates down-forward and then up-backward, while that in the left ear rotates in the inverse direction. The amplitude-modulated echo was simulated by assuming the object has an elevation angle of 30˚.

detection mechanisms have been investigated based on acoustic cues such as ILD [28,29], the Doppler shift effect [20], and hearing directivity deformation by moving the soft material of the ear [30]. The findings indicate that the ear movements employed by bats have the function of obtaining the acoustic cues required for elevation angle detection in the sound pressure domain and/or frequency domain.

Several previous studies have investigated the usefulness of ear motions for echo direction detection through mathematical simulations [29,31] or practical demonstrations [20,32], but the precise nature of effective ear movements remains unclear. Although some studies have measured the precise 3D pinnae motions [20], it would be difficult to understand the optimality of the ear motions by physiological measurements alone. To understand the key components of the ear movements, it is necessary to compare the direction detection performance against that of any other pseudo-movements of the ears.

In contrast, a theoretical approach allows us to evaluate various pinnae motions, including those of bats. Moreover, theoretical investigation can isolate the various factors of acoustic localization and provide insights into their essential components. For example, the ear movements that are suited to individual acoustic cues (i.e., acoustic cues in the sound pressure domain and frequency domain) can be independently considered based on acoustic physics. Thus, theoretical analysis provides the 3D kinematic requirements of pinnae movements for accurate direction detection through binaural listening tasks. The integration of such a pinnae control theory with physiological and behavioral findings may provide an interpretation of bat behavior, and would possibly provide support for biomimetic applications.

Based on these motivations, rigid ear movements for echo source detection based on ILD were mathematically investigated through a series of simulations. The simulations were conducted under the assumption that bats accomplish echo source detection by using the amplitude modulation induced by ear movements in the CF part of the echoes. The distance detection of echo sources with the FM part of the echoes was omitted from our simulations because the distance detection mechanism in bats' echolocation is well known [7,23]. In particular, simulations of different ear motions were analyzed to identify the nature of ear motions that are suited to sound source detection. In these analyses, various ear motions were evaluated in terms of their direction detection performance using custom-made functions and supervised machine learning.

## Methods

### Behavioral traits of bats reflected in our model

In this subsection, we describe the behavioral traits of bats reflected in our model. Fig 1A shows a typical time–frequency structure of the echolocation pulses emitted by CF-FM bats (*Rhinolophus ferrumequinum nippon*) recorded in a previous study [33]. In these pulses, the energy maximum appears in the second harmonic of the CF part ($CF_2$); bats actively use $CF_2$ for fluttering moth detection and Doppler shift compensation [24–26]. To simplify our simulations, amplitude modulation was only calculated for the $CF_2$ component of the echo. Amplitude modulation calculations for the FM part were omitted so that the pure direction detection performance could be evaluated using $CF_2$.

According to previous studies that measured the ear motions of bats, *Rhinolophus ferrumequinum* continuously move their pinnae while listening to the CF part of the echo [20]. These bats adjust their left and right pinnae in an antiphase manner [19,20]. In particular, the pitch angle of the ears tends to move from back to front or from front to back while listening to the echoes. Such movements can be modeled as a cosine phase [19]. Based on these findings, antiphase pitch motions were assumed in the bat-mimicking simulations.

To date, previous measurements have not demonstrated that vertical direction detection is achieved using the amplitude modulation caused by pinnae movement. However, an electrophysiological study reported that the spatial region to which collicular neurons exhibit maximum sensitivity to acoustic stimuli depends on the position of the pinna, and shifts predominantly in the vertical plane when the pinna is retracted manually [28]. According to recent research on acoustic simulation considering the detailed pinnae shape and movements of *Rhinolophus ferrumequinum Nippon*, it was suggested that the pinna movements change the hearing directivity [34]. From the perspective of acoustic physics, it is natural that the amplitude of the echoes is modulated if the hearing directivity changes while receiving the echoes. Therefore, the simulations assumed that the ear movements cause amplitude modulation by shifting the hearing direction. Fig 1B shows a schematic diagram of the $CF_2$ component in the simulated echo assuming that amplitude modulation is caused by the virtual pinnae motion.

The amplitude modulation depth and modulation pattern changes in response to the virtual pinnae motion. The detailed setup for determining the hearing directivity characteristics of the virtual pinnae is described in the next section. Because CF-FM bats tend to conduct the sensing process twice in the space of one periodic pinnae motion [19], echo signals obtained from two sensing operations were simulated in our analyses. With reference to previous measurements of *Rhinolophus ferrumequinum* [20], the echo frequency was set to 70 kHz (i.e., wavelength $\lambda$ = 5 mm) and the echo duration was set to 33 ms. Note that the silent interval does not contribute to the direction detection from a computational perspective. Therefore, we combined the 1st and 2nd echoes in our model, as indicated in Fig 1B.

## Model of the direction detection system

Fig 2A shows a schematic diagram of the virtual environmental setup for the left and right ears and a target object. A single target object was stationed in the direction expressed by the azimuth angle $\theta$ and elevation angle $\varphi$, or equivalently by the unit vector $\boldsymbol{n} = (\cos\theta\cos\varphi, \sin\theta\cos\varphi, \sin\varphi)$, which we call the direction vector. In our model, the amplitude modulation of the echo is caused by changes in the spatial orientation of the virtual ear. Fig 2B shows a schematic diagram of the left and right ears and a speaker when all materials are directed in front of the bat (positive direction of x-axis). In the engineering field, beamforming is a popular method of synthesizing the signals from multiple microphones, as seen in phased array systems [35]. Following this method, four omni-directional microphones were placed at the vertices of a rectangle to construct a virtual ear. The four simulated echo signals obtained from these microphones were summed to generate the overall received signal. In particular, by adjusting the horizontal and vertical spacing between the microphones ($\delta_y$, $\delta_z$), the hearing directivity pattern could be controlled. Fig 2C shows the hearing directivity pattern used in this study. Based on measurements and computational representations of the hearing directivity patterns of CF-FM bats, including *Pteronotus parnellii* [17, 18], *Hipposideros pratti* [30], *Rhinolophus Rouxi* [17], and *Rhinolophus ferrumequinum* [30], the half-amplitude angle (−6 dB off-axis angle from the maximum sensitivity angle) tends to be distributed from 40–90˚

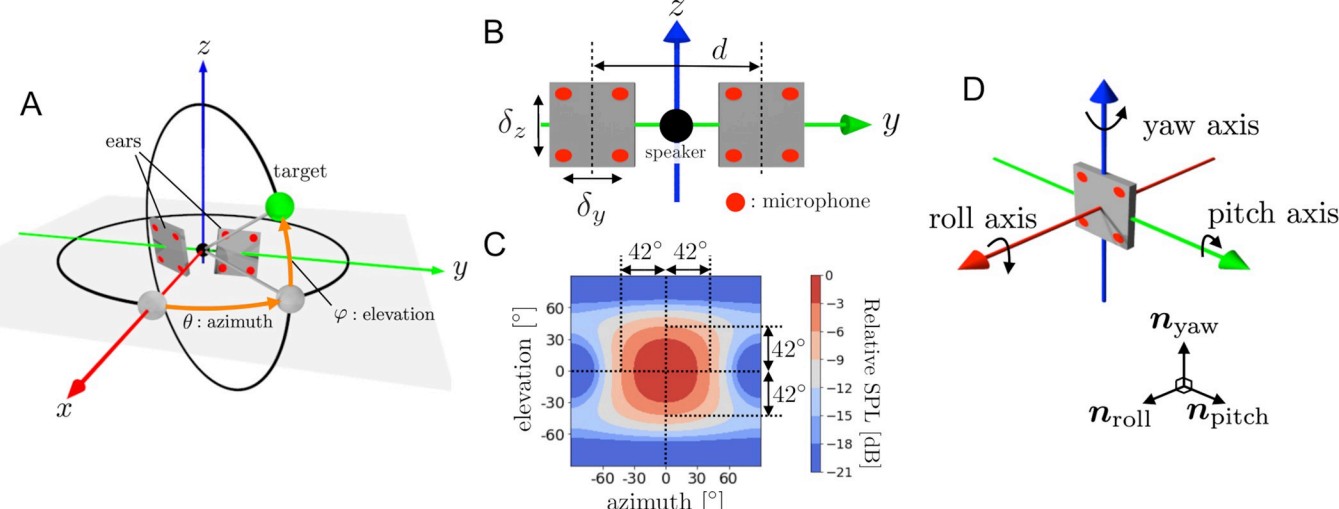

**Fig 2. Schematic diagram of model setup.** (A) Direction of the target (green ball) expressed by azimuth angle $\theta$ and elevation angle $\varphi$. (B) Positions of the two directional ears with spacing $d$. Each red dot indicates the position of an omni-directional microphone. Each ear consists of four omni-directional microphones, where $\delta_y$ and $\delta_z$ are the horizontal and vertical spacings of each microphone. (C) Hearing directivity pattern of the virtual ear. (D) Three axes (roll, pitch, yaw) fixed to the directional ear and corresponding orthonormal basis [$\boldsymbol{n}_{roll}$, $\boldsymbol{n}_{pitch}$, $\boldsymbol{n}_{yaw}$].

off-axis in the horizontal plane. Based on these characteristics, $\delta_y$, $\delta_z$ were set to be slightly smaller than half the echo wavelength $\lambda$. As a result, an anisotropic beampattern was reproduced, as shown in Fig 2C.

As shown in Fig 2D, the roll axis, pitch axis, and yaw axis are fixed to the directional ear and the unit vectors $\boldsymbol{n}_{roll}$, $\boldsymbol{n}_{pitch}$, $\boldsymbol{n}_{yaw}$ indicate the directions of these three axes. The spatial orientation of the directional ear is then given by the matrix $L = [\boldsymbol{n}_{roll}, \boldsymbol{n}_{pitch}, \boldsymbol{n}_{yaw}] \in SO(3)$. Additionally, the orientation change caused by the motion of the directional ear is expressed by the $SO(3)$-valued function $L(t) = [\boldsymbol{n}_{roll}(t), \boldsymbol{n}_{pitch}(t), \boldsymbol{n}_{yaw}(t)]$, where $t$ is the time variable. Assume that the target object is pointed to by the direction vector $\boldsymbol{n}$ and the echo received at the origin is a sinusoidal wave with amplitude $A$ and wavelength $\lambda$. The directional ear in proximity to the origin receives a signal whose envelope component $S_{env}$ is expressed by the following formula (see S1 Text):

$$S_{env}(t; \boldsymbol{n}) = 4A \, \cos\frac{\pi\delta_y\tilde{n}_y(t)}{\lambda} \cos\frac{\pi\delta_z\tilde{n}_z(t)}{\lambda} \tag{1}$$

where $L(t)^T\boldsymbol{n} = \tilde{\boldsymbol{n}}(t) = (\tilde{n}_x(t), \tilde{n}_y(t), \tilde{n}_z(t))$. Therefore, the amplitude modulation of the echo envelope caused by the motion of the directional ear can be calculated for every target direction $\boldsymbol{n}$ once the history of spatial orientation $L(t)$ is known. Note that $L(t)$ and $S_{env}(t; \boldsymbol{n})$ are to be defined for the left and right directional ears. From the envelope of the left and right received echoes, the ILD is defined by the following equation:

$$P(t; \boldsymbol{n}) = 20 \log_{10} \frac{S_{env}^{left}(t; \boldsymbol{n})}{S_{env}^{right}(t; \boldsymbol{n})} \tag{2}$$

where $\boldsymbol{n}$ is the direction vector to the target and $S_{env}^{left}(t; \boldsymbol{n})$, $S_{env}^{right}(t; \boldsymbol{n})$ indicate the envelope of the left and the right received echoes under histories of spatial orientations $L^{left}(t)$ and $L^{right}(t)$, respectively.

The procedure described above obtains the ILD, which is a temporal signal $P(t; \boldsymbol{n})$, from the direction vector $\boldsymbol{n}$. Our question is whether we can obtain the direction vector $\boldsymbol{n}$ from the ILD signal $P(t; \boldsymbol{n})$. If so, what motions of the left and right directional ears make it possible, and how robust is the detection performance to observation errors?

## Evaluation function and degree of injection

In this section, we prepare a general mathematical framework for describing and evaluating active sensing process. The sensing process consists of two phases. In the first phase, we obtain data from the objective system, and in the second phase, we identify the variable of interest based on the obtained data. To evaluate the design of the sensing process, we introduce a general evaluation function and the *degree of injection* index.

Let $X$ be a set of state variables of the objective system, which we are going to identify through the observations. We write the observation process as the map

$$F : \ X \longrightarrow Y \tag{3}$$

where $Y$ is the space in which the observed data lie (possibly a Euclidian space or a functional space). Of course, we can define the map $F$ only when the states of the system having the same state variable of $X$ give the same observation data; hereafter, this is assumed to be true. To determine the state variable uniquely from the observed data, we require the inverse map

$$F^{-1} : \ F(X) \longrightarrow X \tag{4}$$

which gives the computational process. Therefore, the observation map $F$ should be *injective*. In addition, to be sufficiently robust to observation errors, $F$ must be non-degenerate, and hopefully not nearly degenerate at any point in $X$. (Here, 'degenerate' means that the dimension of the tangential map's image is less than the dimension of $X$.) Based on these considerations, we define the evaluation function $U_F$ on $X$ as follows:

$$U_F(x) = \sup_{x' \in X - \{x\}} \frac{d_X(x, x')}{d_Y(F(x), F(x'))} \tag{5}$$

where $d_X$ and $d_Y$ indicate the distance functions defined in spaces $X$ and $Y$, respectively. $U_F(x)$ = $+\infty$ holds if the injective property of $F$ is violated at $x$ (meaning the existence of $x' \neq x$ satisfying $F(x') = F(x)$). In addition, $U_F(x)$ can measure the degree of degeneration of $F$ at $x$. Actually, $U_F(x)$ becomes infinite if $F$ is degenerate at $x$, and it attains a large value if $F$ is nearly degenerate at $x$, which means that the inverse map is too sensitive to observation error at $F(x)$. In any case, the large magnitude of the evaluation function $U_F(x)$ implies difficulty in constructing an inverse map or a well-behaved inverse map at $F(x)$.

Finally, we define the degree of injection of $F$ by the following equation:

$$I[F] = \left(\int_X U_F(x)dx\right)^{-1} \tag{6}$$

Note that $X$ is usually a subset of some Euclidian space, and so the integral is definable. Large values of $I[F]$ indicate that the evaluation function $U_F$ does not take a large value in the state variable space $X$, so the well-behaved inverse map $F^{-1}$ is expected to exist globally. This implies that the observed data contain rich information for determining the desired state variable. Conversely, if $I[F]$ is small, $F^{-1}$ itself or a well-behaved $F^{-1}$ is difficult to construct.

To clarify the meaning of $U_F(x)$ and $I[F]$, we present a full description in the supplemental text (see **S2 Text**).

Our task is to find the direction of the target from the time series data of the ILD. Thus, we consider $X$ as a set of directions expressed by some subset of the unit sphere $S^2$, for example,

$$X = \{\boldsymbol{n} = (\cos\theta\cos\varphi, \sin\theta\cos\varphi, \sin\varphi) \in S^2; |\theta| < \theta_{max}, |\varphi| < \varphi_{max}\} \tag{7}$$

with the 2-norm in $\mathbb{R}^3$. We set the measured data space to $Y = C^0([0, T])$ with the sup-norm, where $T$ is the period of the ear motions. In our problem, the observation process is determined by the spatial orientation change of the left and right directional ears, expressed by the two $SO(3)$-valued functions $L^{left}(t)$ and $L^{right}(t)$ with period $T$. We denote the pair $L^{left}(t)$ and $L^{right}(t)$ as $M$, and use the notation $P_M(t; \boldsymbol{n})$ for the ILD signal obtained by the spatial orientation change $M = (L^{left}(t), L^{right}(t))$. We adopt the same symbol $M$ for the map $M: X \rightarrow Y$ defined by

$$M: \boldsymbol{n} \rightarrow P_M(\cdot; \boldsymbol{n}) \tag{8}$$

Note that the map $M$ is definable because $P_M(\cdot; \boldsymbol{n})$ is a function of the ratio between the amplitudes of the left/right envelope signals, which does not depend on the target distance and other factors like the reflection rate of the object. Following expression (5), we write the evaluation function as

$$U_M(\boldsymbol{n}) = \sup_{\boldsymbol{n}' \in X - \{\boldsymbol{n}\}} \frac{\|\boldsymbol{n} - \boldsymbol{n}'\|_2}{\|P_M(\cdot; \boldsymbol{n}) - P_M(\cdot; \boldsymbol{n}')\|_\infty} \tag{9}$$

and we define the degree of injection of $M$ by

$$I[M] = \left(\int_X U_M(\boldsymbol{n})d\boldsymbol{n}\right)^{-1} = \left(\iint_X U_M(\theta, \varphi)\cos\varphi\, d\theta d\varphi\right)^{-1} \tag{10}$$

Using this index, we will evaluate various types of ear motions and compare them with the quality of the inverse map (pseudo-inverse map in the case of non-injectivity, as discussed later) constructed by the neural network described in the next section. Note that the expression $(\theta, \varphi)$ will often be used instead of the direction vector $\boldsymbol{n}$, as seen in (10), where this will not cause confusion.

## Construction of the inverse map using supervised machine learning

Supervised machine learning is a good tool for constructing an inverse map numerically when an analytical expression is intractable. To confirm that the inverse map can be constructed when the appropriate ear motions are employed, a 3D direction detection test was conducted using a fully connected neural network. Fig 3 shows a schematic diagram of a fully connected neural network and the data flow. Supervised machine learning was performed using this network. The input data to the neural net were the discretized ILD data calculated from the angle pair $(\theta, \varphi)$ under the adopted ear motion $M$, and the output data were the angle pair $(\theta_{guess}, \varphi_{guess})$, i.e., the estimated $(\theta, \varphi)$. The detection error in constructing an inverse map was evaluated by the following equation,

$$E[M] = \max_{(\theta,\varphi)\in X}\left\{|\theta - \theta_{guess}| + |\varphi - \varphi_{guess}|\right\} \tag{11}$$

The azimuth angle $\theta$ and the elevation angle $\varphi$ were restricted within ±60˚. The neural network was trained 5000 times using uniformly distributed random $(\theta, \varphi)$ data. During the last 250 steps of the training, tests were carried out between every training step. In the test condition, the azimuth angle $\theta$ and the elevation angle $\varphi$ were divided into 5.45˚ increments so that

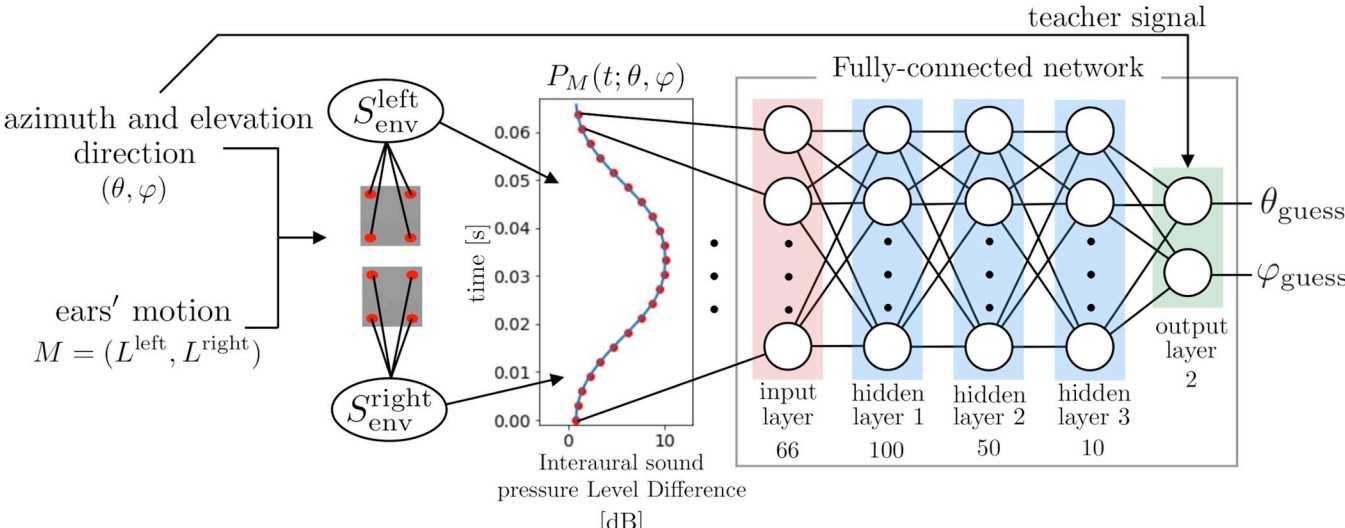

**Fig 3. Schematic diagram of the supervised learning approach for obtaining the inverse map of $M$.** The ILD signal is calculated for all directions $(\theta, \varphi)\in X$ fixing the ear motions. It is discretized at time intervals of 1 ms and passed to the input layer of the neural network. In the neural network, the ReLU activation function is used in hidden layers 1, 2, and 3, and the mean squared error is the error function in the output layer.

23×23 situations were tested, and the detection errors were evaluated for every tested angle pair $(\theta, \varphi)$. Finally, $\theta_{guess}$ and $\varphi_{guess}$ are evaluated as the median of the last 250 output data, respectively.

NOTE1: Under the supervised learning approach described above, the inverse map of $M$ is constructed when $M$ is injective, in some accuracy level. However, our network learns some inverse-like map even when $M$ is not injective, which we call the pseudo-inverse map. This pseudo-inverse map works as follows:

$$P_M(\cdot;\ \theta, \varphi) \rightarrow \text{average of } \{(\theta', \varphi') \in X;\ P_M(\cdot;\ \theta', \varphi') = P_M(\cdot;\ \theta, \varphi)\}, \tag{12}$$

where 'average' means the center of gravity in the $\theta$- $\varphi$ plane in this case.

NOTE 2: Once the inverse map (or pseudo-inverse map) has been constructed, we can evaluate whether it is well-behaved using $E[M]$ in Eq (11). Thus, the degree of injection $I[M]$ appears to be unnecessary. However, the construction of $M^{-1}$ through machine learning is computationally intensive, whereas the degree of injection is relatively simple to calculate. Therefore, considering the computational cost (which is essential in more complicated problems), it is sensible to construct the inverse map $M^{-1}$ only for promising motions $M$ identified by the degree of injection $I[M]$. At the same time, we construct $M^{-1}$ and compare $E[M]$ and $I[M]$ for some typical motions as a means of demonstrating that our new index $I[M]$ works well.

## Setting of directional ear motion patterns

The specific form of the directional ear motions can be written as follows using the roll–pitch–yaw expression (see **S3 Text**):

$$L^{\text{left}}(t) = R_z(\theta_e^l(t))\, R_y(-\varphi_e^l(t))\, R_x(-\psi_e^l(t)) \tag{13}$$

$$L^{\text{right}}(t) = R_z(\theta_e^r(t))\, R_y(-\varphi_e^r(t))\, R_x(-\psi_e^r(t)) \tag{14}$$

where the six angle functions $\psi_e^l$, $\psi_e^r$, $\varphi_e^l$, $\varphi_e^r$, $\theta_e^l$, $\theta_e^r$ are periodic with period $T$, and the frequency of the ear motions is set to $f_e = T^{-1}$. In our model, the periodic motion in each roll, pitch, and yaw component is assumed to be the 0th and 1st Fourier modes. Note that by combining each roll, pitch, and yaw movement, the various ear movements can be represented. Thus, we define the pairing types of the left- and right-ear angle functions as listed in Table 1. In our simulations, the roll, pitch, and yaw angle functions ($\psi_e^{l,r}$, $\varphi_e^{l,r}$ and $\theta_e^{l,r}$) were chosen from the pairing types listed in Table 1.

## Results

### Typical examples for direction detection with ear motions

To confirm the usefulness of the ear motions, two patterns (with and without ear motions) were compared. Fig 4 shows the evaluation function $U_M(\theta, \varphi)$ and the results of machine

**Table 1. Pairing types of left and right angle functions $(\psi_e^l,\ \psi_e^r),\ (\varphi_e^l,\ \varphi_e^r),\ \text{and } (\theta_e^l,\ \theta_e^r)$.**

| Pairing name | Left angle function | Right angle function |
|:---:|:---:|:---:|
| **0** | 0 | 0 |
| $\overline{\textbf{CONST}}$ | $C$ | $-C$ |
| **SIN** | $C\sin(2\pi f_e t)$ | $C\sin(2\pi f_e t)$ |
| $\overline{\textbf{SIN}}$ | $C\sin(2\pi f_e t)$ | $-C\sin(2\pi f_e t)$ |
| **COS** | $C\cos(2\pi f_e t)$ | $C\cos(2\pi f_e t)$ |
| $\overline{\textbf{COS}}$ | $C\cos(2\pi f_e t)$ | $-C\cos(2\pi f_e t)$ |

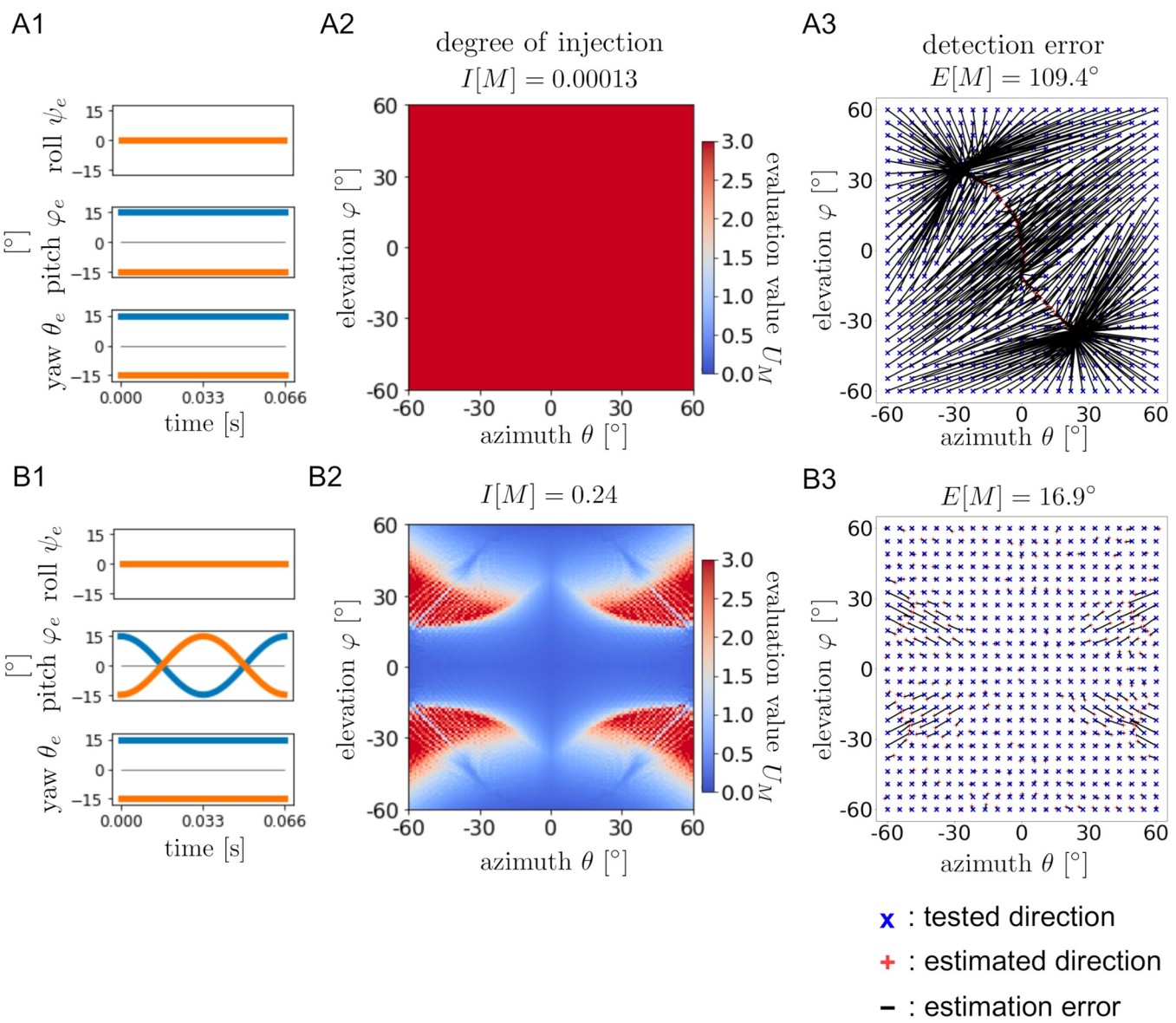

**Fig 4. Examples of direction detection performance with and without ear motions.** (A1, B1): Combination of angle functions. (A2, B2): Colormaps of evaluation function $U_M(\theta, \varphi)$ and the degree of injection $I[M]$. (A3, B3): Results of machine learning. Blue 'x' markers indicate test data $(\theta, \varphi)$ and red '+' markers indicate output data $(\theta_{guess}, \varphi_{guess})$. Black lines are the error lines connecting points $(\theta, \varphi)$ and $(\theta_{guess}, \varphi_{guess})$. The detection error $E[M]$ is also given.

learning under two patterns: $[\psi_e^{l,r}: \mathbf{0}, \varphi_e^{l,r}: \overline{\mathbf{CONST}}, \theta_e^{l,r}: \overline{\mathbf{CONST}}]$ as a static example and $[\psi_e^{l,r}: \mathbf{0},$ $\varphi_e^{l,r}: \overline{\mathbf{COS}}, \theta_e^{l,r}: \overline{\mathbf{CONST}}]$ as a dynamic example. As shown in Fig 4 A2-A3 and B2-B3, the color-map of $U_M(\theta, \varphi)$ reflects the geometric pattern of the distribution of detection errors by the neural network. The degree of injection $I[M]$ is less than 0.001 for the static condition and 0.24 for the dynamic condition. Moreover, the detection error $E[M]$ is 109.4° for the static condition and 16.9° for the dynamic condition.

Examples of more complete direction detection are shown in Fig 5. In these examples, the ear motions conditions were chosen as $[\psi_e^{l,r}: \overline{\mathbf{SIN}}, \varphi_e^{l,r}: \overline{\mathbf{COS}}, \theta_e^{l,r}: \overline{\mathbf{CONST}}]$ and $[\psi_e^{l,r}: \mathbf{SIN}, \varphi_e^{l,r}:$ $\overline{\mathbf{COS}}, \theta_e^{l,r}: \overline{\mathbf{SIN}}]$. In each condition, the evaluation function $U_M(\theta, \varphi)$ takes smaller values in the whole domain, and the degrees of injection $I[M]$ are 1.52 and 1.35, respectively. The detection

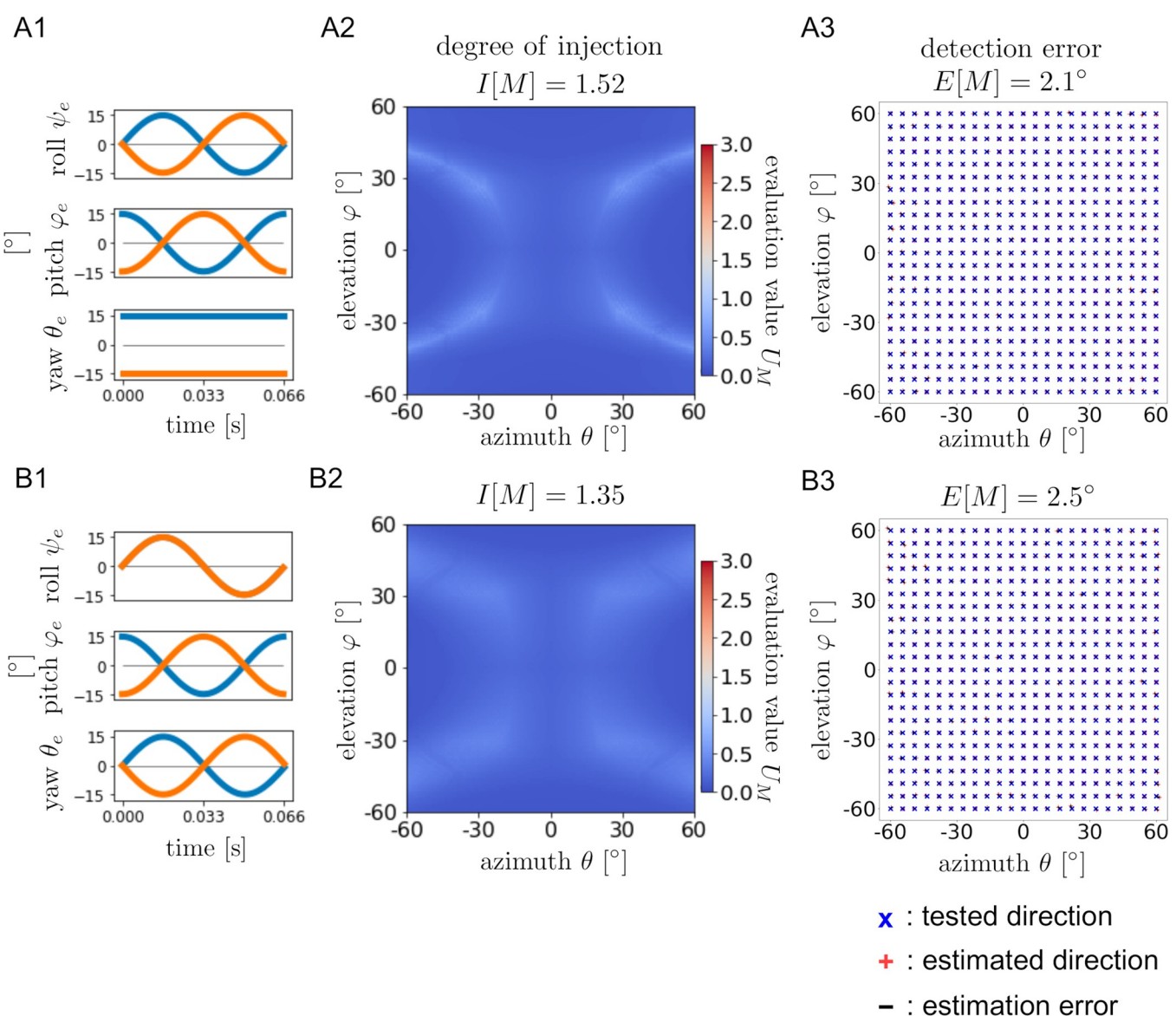

**Fig 5. Examples of direction detection performance with appropriate ear motions.** The formation of Fig 5 is same as Fig 4. Blue color map and less-visible error lines mean the good performance of direction detection.

errors $E[M]$ are 2.1˚ and 2.5˚, indicating that relatively accurate direction detection is accomplished compared with the previous examples shown in Fig 4. These results suggest that it is necessary to combine the roll, pitch, and yaw rotations appropriately for accurate detection of the direction. Additionally, the results in Figs 4 and 5 indicate that the degree of injection is strongly related to the direction detection performance.

## Exhaustive analysis of ear motions in pitch anti-phase case

To determine appropriate combinations of the roll, pitch, and yaw rotations, 36 motion patterns were analyzed. The corresponding evaluation functions and degrees of injection are shown in Fig 6. As described before, based on the actual motions of bats' pinnae, the pitch angle functions $\varphi_e^{l,r}$ are fixed to the anti-phase pairing pattern $\overline{\textbf{COS}}$.

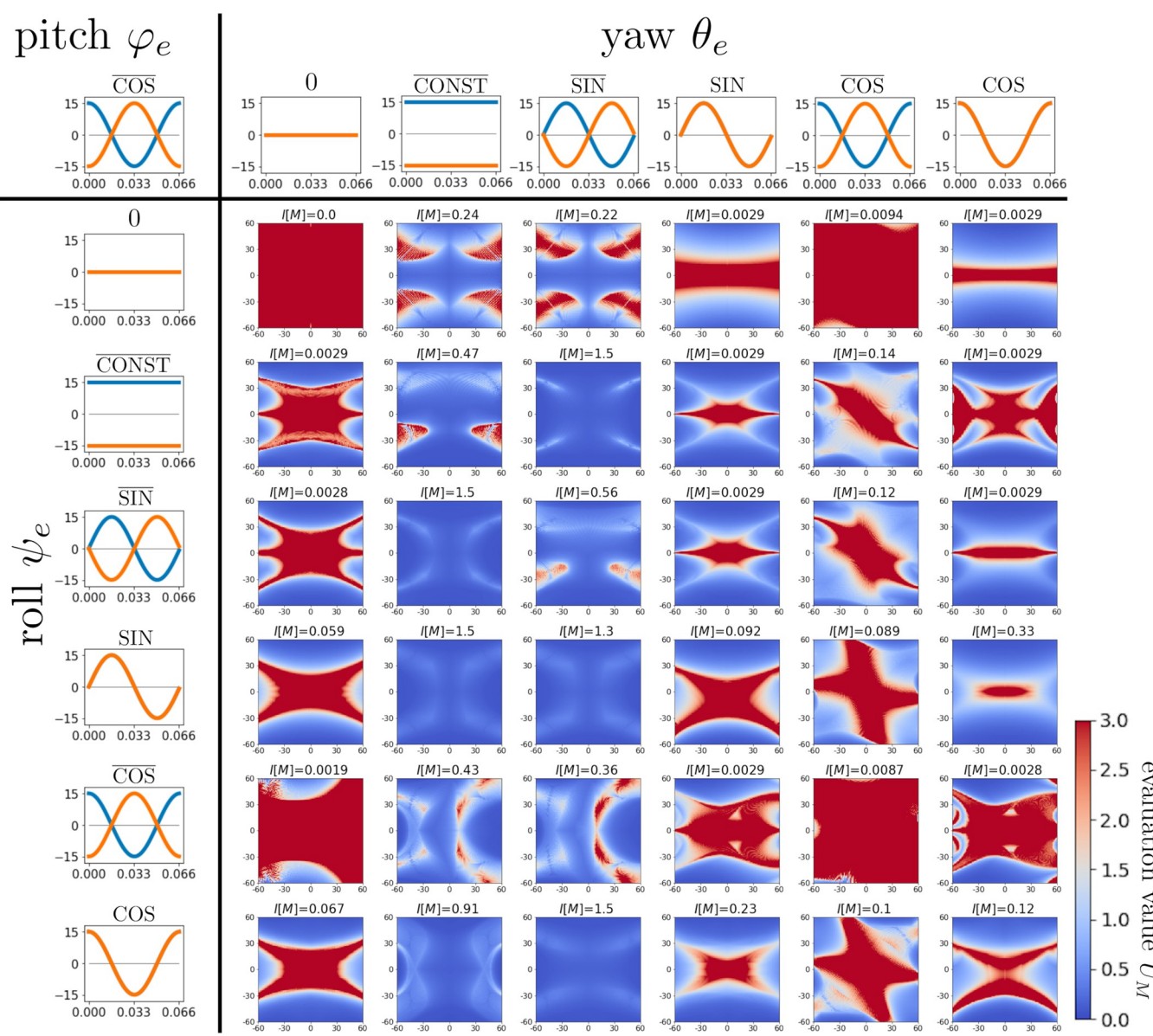

**Fig 6. Colormaps of $U_M(\theta, \varphi)$ and the degrees of injection for various ear motion patterns.** The pitch angle functions $\varphi_e^{l,r}$ are fixed to $\overline{\text{COS}}$ according to actual bat behavior. Blue and orange lines indicate the angle functions of the left and right ears, respectively. The left and top array panels display the roll angle functions $\psi_e^{l,r}$ and the yaw angle functions $\theta_e^{l,r}$, respectively.

To classify the ear motion patterns graphically, we focus on the orbits of ear motions given by $(\psi_e^l(t), \varphi_e^l(t), \theta_e^l(t))$ and $(\psi_e^r(t), \varphi_e^r(t), \theta_e^r(t))$ in $\psi_e-\varphi_e-\theta_e$ space. Additionally, the convex hull of the union of the left and right ears' orbits in $\psi_e-\varphi_e-\theta_e$ space is considered. We classify the motion patterns according to the pair of dimensions of the convex hull and each ear's orbit. As shown in Fig 7, there are five types of dimension pairs: 3–2, 3–1, 2–2, 2–1, and 1–1.

Fig 8 exhibits the dimension pairs of the convex hull and each ear's orbit, the degree of injection $I[M]$, and the detection errors $E[M]$ of the 36 motion patterns. If we assume that $E[M]<5°$ is the criterion for precise direction detection, there are 12 motion patterns (colored boxes) that achieve this level of accuracy. Among them, five patterns (boxes bounded by red

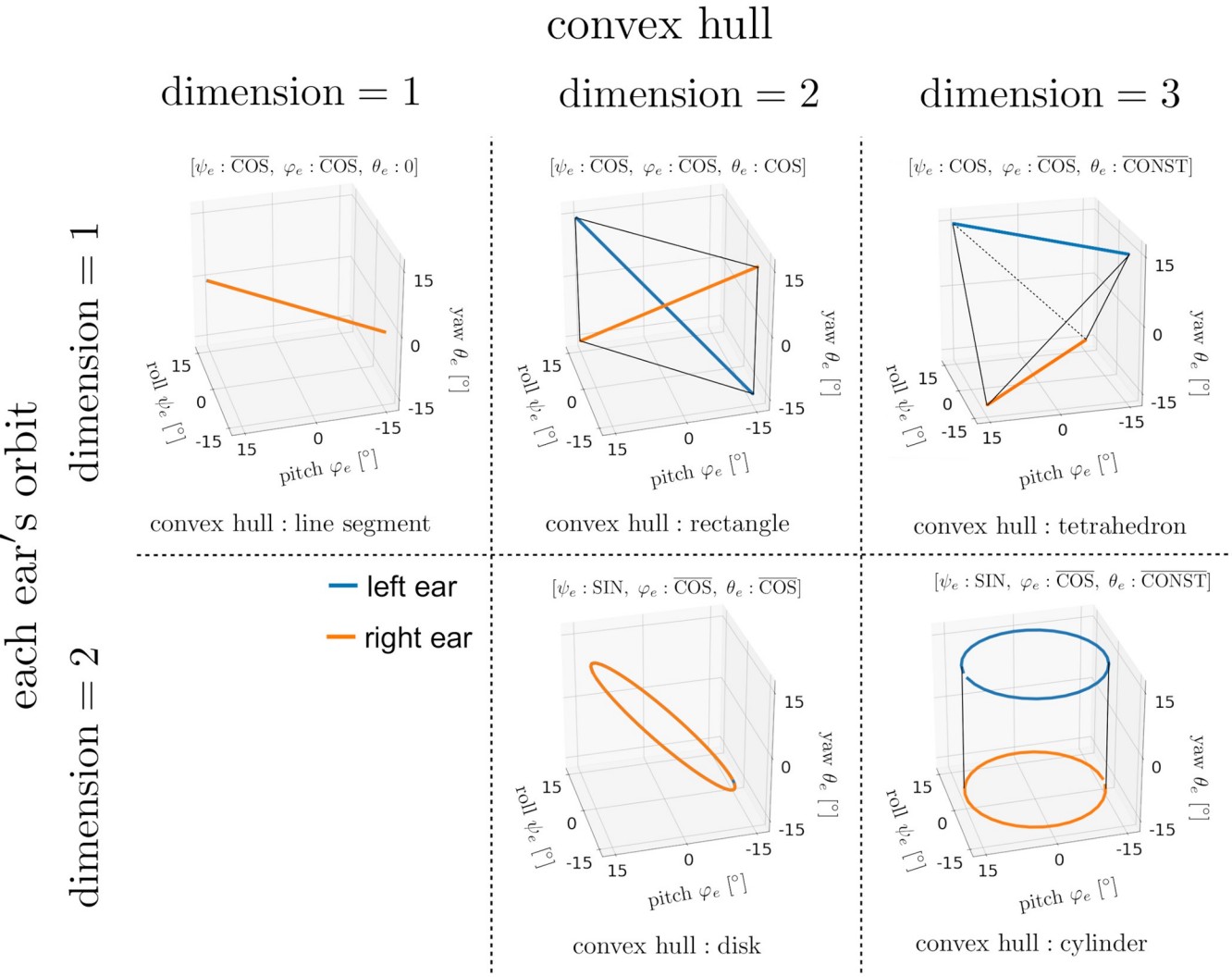

**Fig 7. Five types of dimension pairs of the convex hull and each ear's orbit.** The blue lines indicate the left ear's orbit $(\psi_e^l(t), \varphi_e^l(t), \theta_e^l(t))$ and the orange lines indicate the right ear's orbit $(\psi_e^r(t), \varphi_e^r(t), \theta_e^r(t))$. When both orbits coincide, only the orange line is displayed. The convex hull of the union of both ears' orbits is displayed in each case.

lines) have large injection degrees, essentially indicating robust direction detection motion patterns, as discussed in the next subsection.

## Robustness against degradation of the ILD resolution

Next, the robustness of direction detection against the degradation of the ILD resolution was investigated. Fig 9A shows the relationship between the degree of injection and the direction detection error for the 36 ear motion patterns without the degradation of the ILD resolution (see green line in Fig 9B). We examined the detection robustness against the degradation of the ILD resolution for the relatively small detection error group (i.e., $E[M] < 20°$). The detection errors were reevaluated by decreasing the ILD resolution to 1 dB and 3 dB (see the orange and blue lines in Fig 9B). As shown in Fig 9C, the detection errors remained small for the group with larger degrees of injection ($I[M]>1$), while the errors increased much more in the

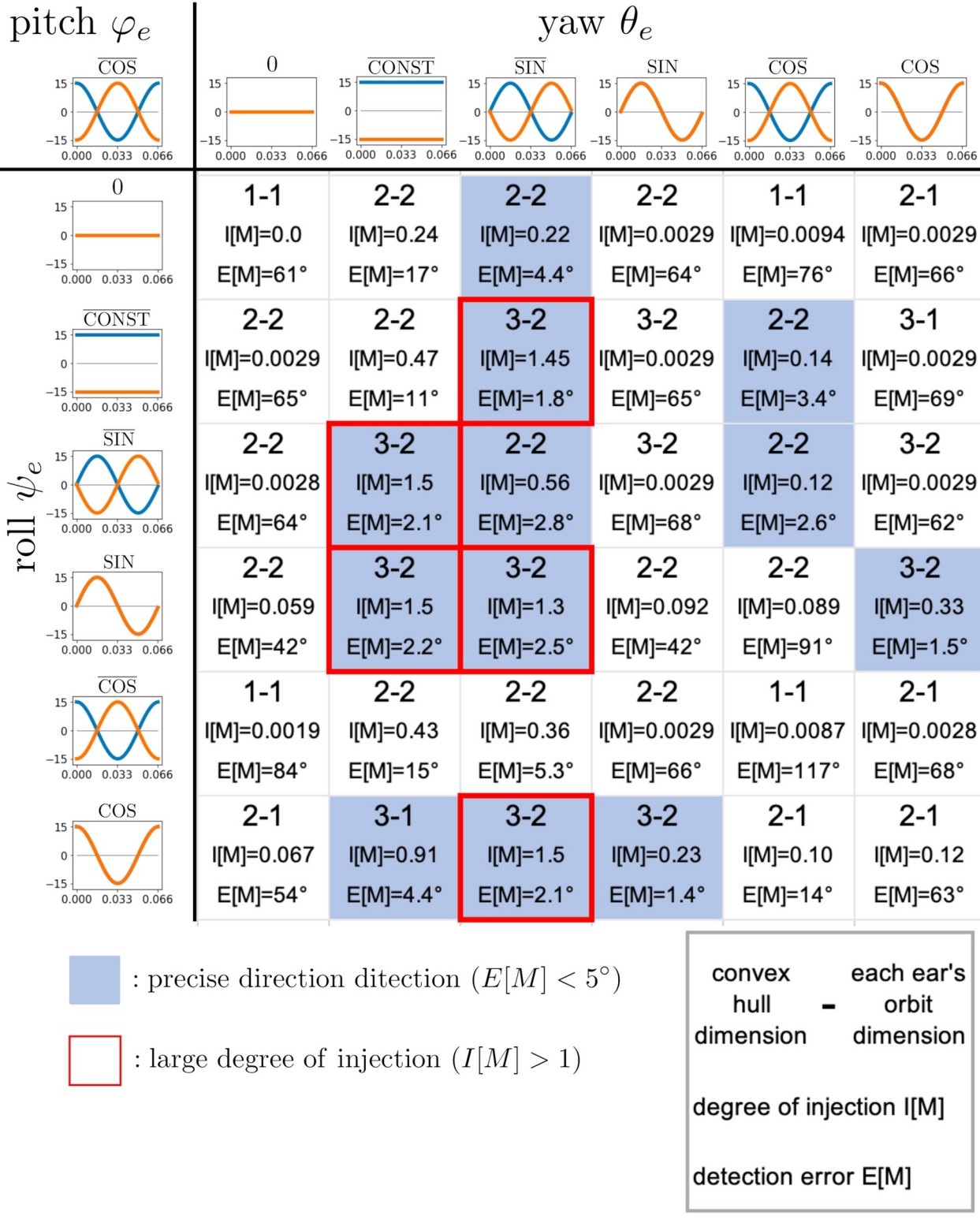

**Fig 8. Dimension pairs and direction detection errors for various motion patterns.** In each box, the dimension pair of the convex hull and each ear's orbit is given in the upper part, the degree of injection is given in the middle, and the detection error is given at the bottom. Here, we adopt $E[M]<5°$ as the criterion for precise direction detection. The colored boxes indicate that the corresponding motion patterns give precise direction detection. The boxes bounded by red lines correspond to the motion patterns with large degrees of injection ($I[M]>1$).

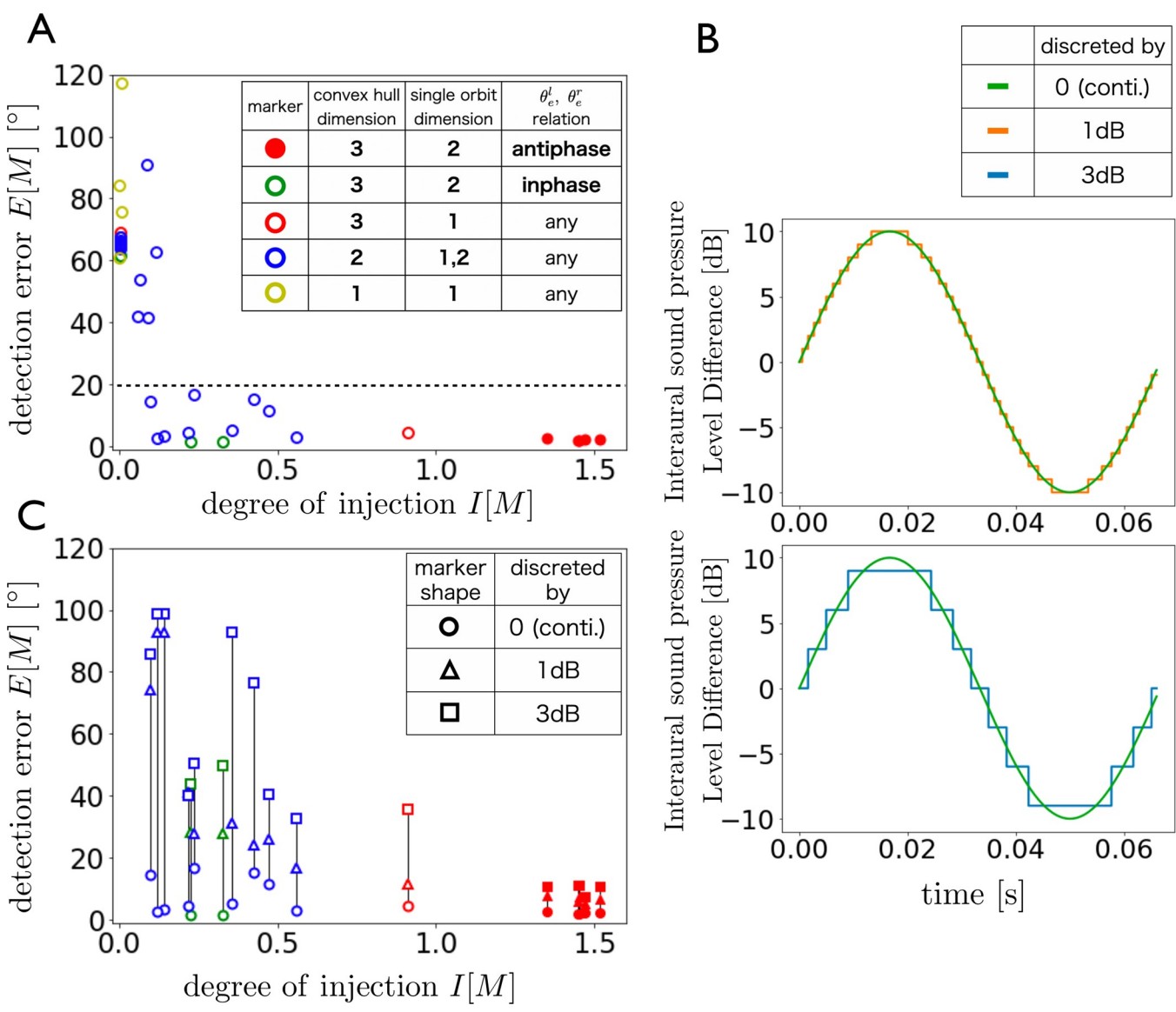

**Fig 9. Relationship between $I[M]$ and $E[M]$ under various degradation levels of ILD resolution.** (A) Relationship between the degree of injection and the detection error for the 36 ear motions without the degradation of the ILD resolution. (B) Example of change in the sinusoidal signal for each degradation level. (C) Relationship between the degree of injection and the detection error for each ear motion under the degraded ILD resolutions. Note that these evaluations were conducted for ear motions with relatively small detection errors ($E[M]<20°$) in the no degradation condition (A). The length of the vertical black line corresponds to the increase in the detection error when the ILD discretization level changes from 0 dB to 3 dB.

other groups. These findings suggest that 5 motion patterns satisfying conditions $I[M]>1$ not only accomplish accurate direction detection, but are also robust to the degradation of the ILD resolution. From these characteristics and Fig 8, we can identify three ear motion conditions that ensure the precise and robust direction detection:

i. The convex hull of the union of the two ear orbits is three-dimensional;

ii. Neither orbit degenerates to one dimension;

iii. The left and right yaw angle functions do not coincide.

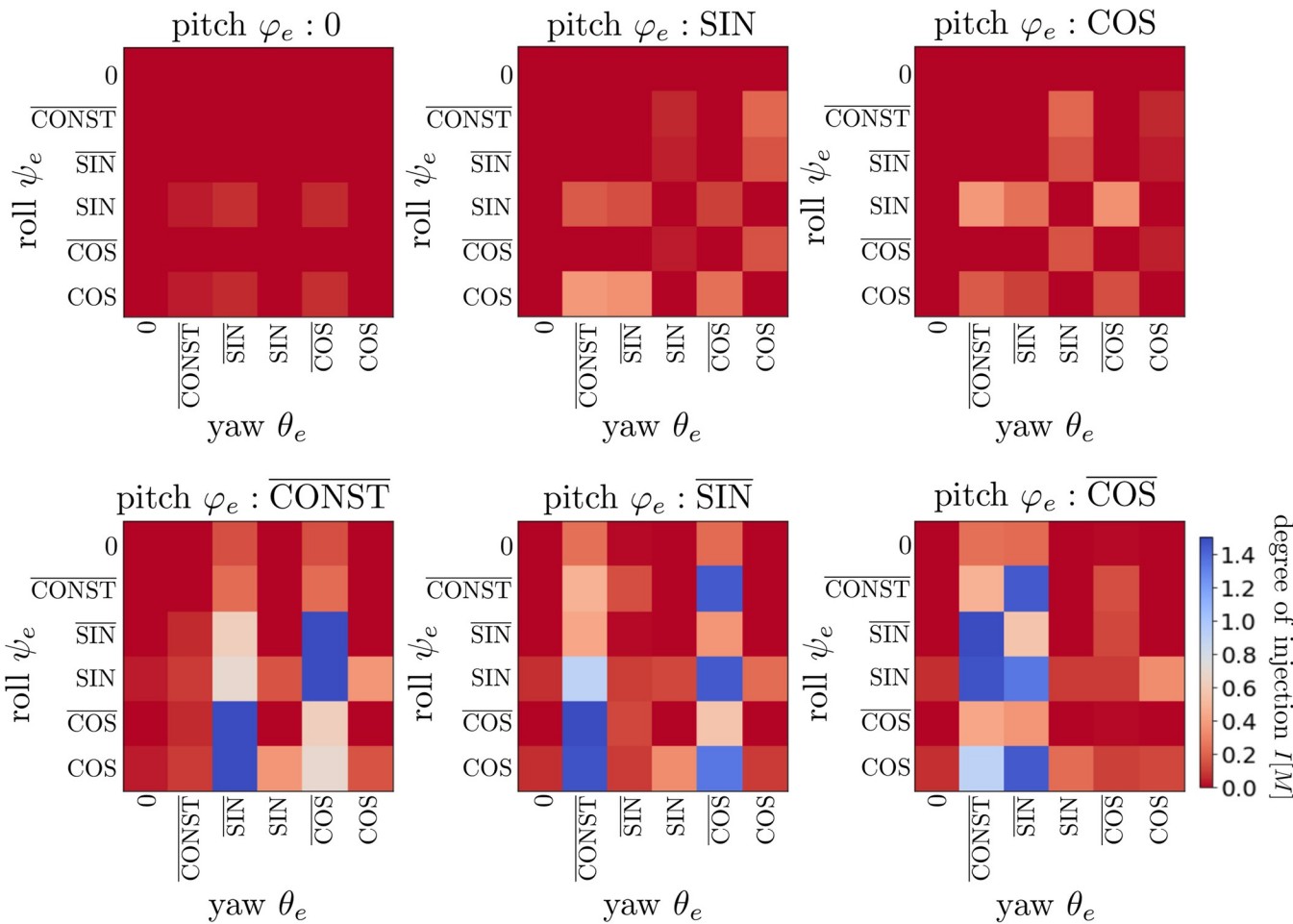

**Fig 10. Colormaps of degree of injection $I[M]$ of all combinations of $\psi_e$-$\varphi_e$-$\theta_e$ angle functions.** The fixation of the pitch angle functions $\varphi_e^{l,r}$ to $\overline{\text{COS}}$ is removed, so that the degrees of injection were evaluated for $6^3 = 216$ motion patterns.

## General case analysis

We now examine the general case. By removing the bat-motivated limitation of pitch motion ($\varphi_e^{l,r}$: $\overline{\text{COS}}$), the detection performances were evaluated for $6^3 = 216$ ear motions in terms of the degree of injection $I[M]$, as shown in Fig 10. These analyses show that the degree of injection $I[M]$ is small when the angle relations $\theta_e^l \equiv \theta_e^r$ OR $\varphi_e^l \equiv \varphi_e^r$ hold. Through these analyses, we determined the following conditions for ear motions satisfying $I[M]>1$:

i. The convex hull of the union of the two ear orbits is three-dimensional;

ii. Neither orbit degenerates to one dimension;

iii. The left and right yaw angle functions do not coincide;

iv. The left and right pitch angle functions do not coincide.

The 14 of 216 motion patterns satisfy the above four conditions. We confirmed that these 14 motion patterns achieve the precise and robust direction detection, and the other patterns do not.

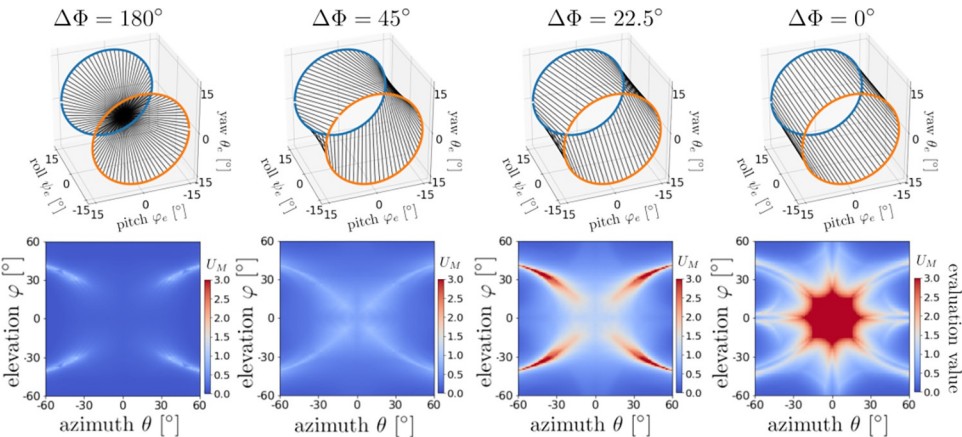

**Fig 11. Effect of phase difference of ear motions on direction detection performance.** In the upper panels, blue line indicates the left ear's orbit given by $(\psi_e^l, \varphi_e^l, \theta_e^l) = (C, C\cos(2\pi f_e t), C\sin(2\pi f_e t))$, and orange line does the right ear's orbit $(\psi_e^r, \varphi_e^r, \theta_e^r) = (-C, C\cos(2\pi f_e t + \Delta\Phi), C\sin(2\pi f_e t + \Delta\Phi))$, respectively. Black straight lines connect simultaneous points of the left and right ears' orbits with the phase difference $\Delta\Phi$. In particular, the ear motion with $\Delta\Phi = 180°$ is $[\overline{\text{CONST}}, \overline{\text{COS}}, \overline{\text{SIN}}]$ and the ear motion with $\Delta\Phi = 0°$ is $[\overline{\text{CONST}}, \text{COS}, \text{SIN}]$. For motions with $\Delta\Phi$ between 45° and 180°, good-quality direction detection performance is achieved.

Finally, the effect of phase differences in the left and right ear motions on the detection performance is examined in Fig 11. All motions have the same orbits, but the simultaneous lines vary according to the pitch–yaw ($\varphi_e$-$\theta_e$) phase difference. This result suggests that phase differences larger than several tens of degrees is sufficient to achieve good-quality detection.

## Discussion

In this study, mathematical models of rigid ear movements for direction detection with ILD were investigated by assuming that ear movements create useful amplitude modulations in the echoes. Our theoretical investigation clarified that only certain simulated ear motions can achieve echo source direction detection accurately (Figs 8 and 10) and robustly (Fig 9).

An investigation based on precise measurements of the pinnae motions and pinnae structure is critical for understanding the direction detection mechanism of certain bat species. In particular, Yin & Müller measured the precise pinnae motions of *Rhinolophus ferrumequinum* [20] and demonstrated the spatial perception system by an artificial mimicking approach [36]. From the general perspective of acoustic sensing, the fundamental question is "what kind of pinnae structure and what kind of pinnae motions have a critical effect on direction detection?" To extract the essential requirements for the pinnae structure and motion, we believe that it is also important to accumulate theoretical knowledge from simplified design simulations. In particular, by assuming that ear movements create useful amplitude modulations in the echoes, our study has clarified the conditions for the precise nature in common with better movement patterns for direction detection through comparisons of various simulated ear movements.

One of our important messages is that, in the case of a rigid ear that reproduces the simple anisotropic hearing directivity pattern, accurate direction detection is theoretically guaranteed if the ear movements satisfy the four conditions described in this paper. This is an important message from theoretical scientists to behavioral scientists, and is more important than the five specific simulated ear movements identified as less detection error movement patterns. For example, our study reproduced ear rotation by assuming sine and cosine functions, but actual bats are unlikely to employ such coherent functional rotation. Our generalized message does not force such a coherent rotation. Any movement is acceptable, as long as it satisfies the four

conditions. The suggestion of such a generalized movement is one of the advantages of our theoretical approach. We believe that such a theoretical analysis is useful for supporting a behavioral understanding because generalized theoretical requirements accept the degree of freedom necessary for ear movements.

In our simulations, a simple anisotropic directivity pattern was assumed for hearing. However, actual measurements of bats have shown that the hearing directivity pattern is more complex in periphery regions [17,18,30,37]. Another study showed that the hearing directivity pattern of bats is deformed by the bending of the soft material of the ears [30]. Such a complex hearing directivity pattern has not been investigated in the present study. It may be possible to deregulate the ear motion conditions by implementing more precise hearing directivity patterns. If so, this would provide a better understanding of the directivity formation effect in bats' pinnae. From another perspective, it has been suggested that the ear movements produce detectable Doppler shifts, with the time–frequency Doppler shift signatures encoding the sound source direction in an orderly fashion [20]. If we were able to clarify the necessary ear movement conditions for direction detection with Doppler shift signatures using a similar analysis framework, we could compare the differences in motion conditions between ILD-based and Doppler shift-based direction detection. Such a comparison could be expected to advance the discussion about the true direction detection mechanism employed by bats.

Previous mathematical studies [29,31] and practical demonstrations [32] have shown that ear motions can be useful under certain motion patterns. In contrast, our study has considered the theoretical basis for these ear motions by evaluating exhaustive simulated motion patterns. Thus, this is the first article to investigate the underlying theory behind the ear motion strategies. According to a behavioral experiment conducted on *Rhinolophus ferrumequinum*, when the ears are fixed so that no movements can be performed, the bats are still able to localize the azimuth of the target positions, but not the elevation [27]. Our simulations under the no-ear-movement condition confirmed similar direction detection performance (see **S1 Fig**), suggesting that continuous ear movements contribute to the elevation angle detection. The results of general case analyses (Fig 10) show that three-axis rotations are necessary for azimuth and elevation direction detection (i.e., those not including the pairing name **0**). However, all three axes do not necessarily need to rotate continuously in time, so two-axis temporal rotation patterns were also included in the movement patterns that accomplish accurate direction detection (i.e., cases including the pairing name $\overline{\textbf{CONST}}$).

In particular, the pitch angle functions $\varphi_e^l$ and $\varphi_e^r$ must retain a different phase, as shown in Fig 10. Such antiphase control of pitch motions has been observed in bats [19], and so our theory strongly supports the inevitability of pitch control in actual bat behavior. Our analyses indicate that the same antiphase control restriction exists in the yaw angle functions $\theta_e^l$ and $\theta_e^r$, but the roll angle functions $\psi_e^l$ and $\psi_e^r$ have no such restriction. These differences might be caused by the fact that the pitch and yaw angles determine the central direction of the directivity pattern, while the roll angle determines the rotation around the direction axis. Thus, our investigations provide not only theoretical support for bats' behavior, but also a new interpretation for roll–pitch–yaw control. Such a theoretical consideration is expected to provide a useful control design for the artificial ear motions of future active listening systems.

Ear motions that give accurate and robust direction detection were only found in five of the 36 motion patterns analyzed in this study, as shown in Figs 8 and 9. It is plausible that the motions of the left and right ears are mirror-symmetric with respect to the midsagittal plane. If so, the following equations should hold:

$$\psi_e^l(t + T/2) = -\psi_e^r(t), \ \varphi_e^l(t + T/2) = \varphi_e^r(t), \ \theta_e^l(t + T/2) = -\theta_e^r(t) \tag{15}$$

Only one of the five high-performance patterns satisfies the above equations, namely [**SIN,** $\overline{\textbf{COS}}$, $\overline{\textbf{CONST}}$] (see the animation in **S1 Video**). This pattern may be easier to implement in artificial active listening systems using ILD codes than the other four patterns.

We have not only identified a wide array of patterns that produce appropriate ear motions (Fig 10), but have also provided simple discrimination conditions using orbits in the roll–pitch–yaw space. Our graph-based evaluation method is also useful for ethological investigation, because the graphs can be drawn using actual measurement data. Moreover, the hearing directivity pattern can be approximated from actual measurement data. Thus, we have not only presented theoretical findings, but also provided an extendable framework of theoretical analysis for ethological research.

In our study, only one directionality of the ear and limited numbers of ear motions were investigated; thus, our theory is not universal. However, the significance of this study lies in showing that simple hearing directionality shapes and well-selected, uncomplicated ear motions are sufficient to achieve precise and robust direction detection. In addition, we proposed an index (degree of injection) that can judge whether the well-behaved inverse map is constructible or not using only the original map, without requiring the construction of an inverse map. Thus, we expect our theory to be useful for general-purpose evaluation systems in sensing fields.

There are still many things to consider about the direction detection mechanism employed by bats. For example, the direction detection mechanism for fluttering prey is important in understanding more practical situations for hunting bats. The acoustic glint generated in the CF component by a fluttering moth [38] may have a positive influence on direction detection, but we have not yet performed any useful investigations to clarify this. Investigations including the acoustic glint would provide more useful insights into the direction detection mechanism with spectral cues and with ILD. In another perspective for future work, bat species who use only broadband FM signals accomplish direction detection without fast ear movements. For these echolocating bats, the spectral filtering function due to the interference of the echoes on the pinnae and tragus surfaces are regarded as the most important feature because ultrasound of various wavelengths provides rich information of every spectral strength [30,39]. However, we have not yet investigated the direction detection mechanism based on spectral cues.

We must also consider the difference between animal intelligence and artificial intelligence. The process whereby animals acquire spatial perception is an important theme. The number of trainings for acquiring the spatial perception on artificial intelligence seems to be much greater than that for animal intelligence. In addition, it is more difficult to identify a supervisor in the case of actual animal learning. Understanding the generalization capabilities of living organisms will bridge the gap to artificial intelligence. Although supervised machine learning is used as an inverse map generation tool in this study, we will investigate the ability to generalize the whole direction detection mechanism by learning a subset of directional detections in a future study.

## Supporting information

**S1 Video. Example movie for appropriate motion of left and right ears.**
(MP4)

**S1 Text. Echo amplitude representation procedure with four omni-directional microphones.**
(PDF)

**S2 Text. Explanations of the evaluation function U$_F$ and degree of injection I[F].**
(PDF)

**S3 Text. Expression of spatial orientation of the directional ear.**
(PDF)

**S1 Fig. Examples of the direction detection performance without ear motion using supervised machine learning.** The ear motion condition was chosen as $[\psi_e^{l,r}: \mathbf{0}, \varphi_e^{l,r}: \mathbf{0}, \theta_e^{l,r}: \overline{\mathbf{CONST}}]$. Blue 'x' markers indicate test data ($\theta$, $\varphi$) and red '+' markers indicate output data ($\theta_{guess}$, $\varphi_{guess}$). Black lines denote the error lines connecting points ($\theta$, $\varphi$) and ($\theta_{guess}$, $\varphi_{guess}$). Each detection error line tends to stretch vertically, indicating that the elevation angle is difficult to detect while the azimuth angle can be accurately detected.
(TIF)

## Acknowledgments

We are grateful to Toshihira Mishima for useful discussions and for providing useful insights that will serve as seeds for my research. We thank Stuart Jenkinson, PhD, from Edanz (https://jp.edanz.com/ac) for editing a draft of this manuscript.

## Author Contributions

**Conceptualization:** Yasufumi Yamada.

**Methodology:** Ryo Kobayashi.

**Software:** Takahiro Hiraga.

**Writing – original draft:** Takahiro Hiraga.

**Writing – review & editing:** Yasufumi Yamada, Ryo Kobayashi.

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
