## [Decision Letter · Decision Letter 0]

22 Feb 2022

Dear Dr. Yamada,

Thank you very much for submitting your manuscript "Theoretical investigation of active listening behavior based on the echolocation of CF-FM bats" for consideration at PLOS Computational Biology.

As with all papers reviewed by the journal, your manuscript was reviewed by members of the editorial board and by several independent reviewers. In light of the reviews (below this email), we would like to invite the resubmission of a significantly-revised version that takes into account the reviewers' comments.

The reviewers are agreed that the paper is of interest, and appropriate for PLOS Computational Biology, but have substantial required changes. I therefore agree with their recommendation to request Major Revisions. Please pay attention to the reviewers' feedback. This includes fixing the data availability issue.

We cannot make any decision about publication until we have seen the revised manuscript and your response to the reviewers' comments. Your revised manuscript is also likely to be sent to reviewers for further evaluation.

Sincerely,

Dan Stowell

Associate Editor

PLOS Computational Biology

Natalia Komarova

Deputy Editor

PLOS Computational Biology

The reviewers are agreed that the paper is of interest, and appropriate for PLOS Computational Biology, but have substantial required changes. I therefore agree with their recommendation to request Major Revisions. Please pay attention to the reviewers' feedback. This includes fixing the data availability issue.

Reviewer's Responses to Questions

**Comments to the Authors:**

Reviewer #1: Hiraga et al. present an interesting theoretical study analyzing the benefits of ear movements on sound localization. The role of ear movements in sound localization is still not fully understood and this study takes us a step forward by testing different movement combinations. One of the interesting findings in the paper is that the pitch movement of the two ears should be in anti-phase which is what is observed in many bats. The study thus makes an important contribution to the question but unfortunately, the manuscript was very difficult to understand. Most of my comments below arise from a failure to understand the authors. I will give several examples:

The authors provide a long section about general estimation in the methods titled: evaluation function and degree of injection, but it was very difficult for me to read how this method connects to the machine learning and why are there two different methods of evaluation.

Moreover, I did not understand that use of ‘X’ or ‘n’ in the injection function. Shouldn’t it rely on Y only? Aren’t we aiming to predict similarity in x based on similarity in y? I feel that there is much to clarify

I also did not understand why are the microphones placed in 4 discrete locations if this is a simulation? Why are they not continuous?

L128 ‘Based on these findings, asymmetrical ear motions were embedded in bat mimicking

Simulations’ – perhaps it is a problem of the English but it is not clear what do the authors mean by asymmetric? Do they mean anti-phase?

I had two scientific comments:

1) The model completely ignores the spectral filtering that is happening in the ear. What about the HRTF? And what about moving / fluttering prey which is what these bats typically hunt?

2) Would be interesting to train the system on part of the region and test it on the rest. How animals generalize during learning is a fundamental question.

There are also many places where the English should be corrected. Here are just a few examples (I could not mark them all). I strongly suggest editing the manuscript by a native English-speaking editor.

English - L72. tasks in the air,

L80 - The lateral superior olive in bats has a relatively large capacity

L103 - bat behavior always exhibits the best-benefit response.

L150 – altitude ?

Finally:

Figure legends are too short making it impossible to understand the figures.

This paper should be read and cited – fast moving bat ears create information Doppler shifts (by Mueller et al.)

Reviewer #2: Review of PCOMPBIOL-D-21-02303 by Hiraga et al.

In this study, the authors used a theoretical approach to investigate which type of outer ear movements can facilitate sound source localization. They modeled an active listening system based on simulated pinna movements and artificial echolocation signals of a CF-bat, Rhinolophus ferrumequinum, and found that only specific three-axis rotational movements of the pinnae are able to facilitate accurate sound source localization in the frontal hemisphere. While the findings presented in the manuscript are generally interesting and might have the potential to advance the field of active acoustic sensing, their relevance for and applicability to natural behavior is unclear.

Despite the model being based on pure artificial input, the authors directly relate the model’s output to the natural behavior of bats. In my eyes, this is only valid if it can be shown that the simulated ear movements implemented in the model resemble those measured in a natural situation. What is the advantage in knowing which type of ear movement allows for maximal sound localization accuracy when it is not known if the bats are physiologically able to perform this kind of ear movement? If it is not possible for the authors to validate their findings by experimental data, I recommend to remove or at least to weaken the statements about the behavioral relevance of their findings throughout the manuscript.

The authors claim that with their study, they evaluated the importance of ear movements for 3D target localization behavior in echolocating bats. It is widely accepted that bats estimate the distance of a target (i.e. target position in the depth dimension) from the temporal delay between the emitted call and the returning echo. Information on echo delay was, however, not implemented into the model in the present study, and therefore the authors’ claim is not valid. The study only allows to draw conclusions regarding target localization in 2D (azimuth/elevation).

The model generated by the authors is based on the assumption that pinna movements in bats produce a certain pattern of interaural level differences (ILDs), and that these ILDs are responsible for 3D target localization. 1) While it seems indeed to be widely assumed that pinna movements generate amplitude modulations in echoes arriving at a bat’s ears, and consequently a temporal pattern of ILDs, I am wondering whether there exists experimental evidence for this assumption. Measurements of natural ear movements in R. ferrumequinum, and of the effects these movements have on sounds arriving at the bat’s ear canal, have so far only shown that the ear movements produce detectable Doppler shifts, and that the time-frequency Doppler shift signatures at the bat’s ear drum encode sound source direction in an orderly fashion (Yin & Müller, 2019, Proc Natl Acad Sci U S A 116(25):12270-12274). 2) It has often been demonstrated that bats use binaural acoustic cues such as ILDs for sound localization in the horizontal plane. For sound localization in the vertical plane, however, spectral sound properties, such as notches in the echo spectrum generated by the sound filtering properties of the outer ears, are used, and to estimate target distance, bats use the time delay between emitted call and returning echo. So, what is the background for assuming that ILDs can enable target localization in elevation and in the depth dimension? Interestingly, when the ears of R. ferrumeqinum are fixed so that no ear movements could be performed by the bat, the bat is still able to localize target positions in azimuth, but not in elevation (Mogdans et al., 1988, The Journal of the Acoustical Society of America 84, 1676-1679). This indicates that ear movements in this bat may indeed be beneficial for vertical target localization. However, the underlying mechanism is unclear. It might well be possible that the bats indeed use the direction dependent Doppler shift signatures for sound localization in the vertical plane.

Abstract: It should be clearly communicated in the abstract that in this study the effect of simulated and not natural ear movements on target localization were investigated. Furthermore, “3D” should be replaced with “2D” throughout the abstract except for the first sentence.

Line 33: This sentence is confusing. To detect a sound source, it is not necessary to know its distance and direction. However, this information is vital to determine the location of the sound source. I recommend to replace “detect” by “determine”.

Line 35: The statement that ear movements enable 3D target localization is false. Ear movements are only helpful in CF bats to localize a sound source in elevation. Horizontal target localization is well possible without ear movements. Please see Mogdans et al., 1988, The Journal of the Acoustical Society of America 84, 1676-1679. Furthermore, target distance estimation is based on pulse-echo delays and should be entirely independent from ear movements.

Line 54: What is meant with “essential theory behind ear movements”? Do the authors refer to the mechanism or the effect of the movement?

Line 59: The significance of pinna movements for echolocation has been revealed already some time ago, e.g. by Mogdans et al., 1988, The Journal of the Acoustical Society of America 84, 1676-1679.

Line 69: There might be a better definition of ECHOLOCATION. For example, echolocation does not necessarily require ultrasound.

Line 81: ILDs are indeed important for horizontal target localization in bats, but for target localization in the vertical plane, the sound filtering properties of outer ear structures are exploited (see e.g. Lawrence & Simmons, 1982, Science 218:481-483). And for target distance estimation, bats use call-echo delays. So, 3D sound localization in bats is based on a combination of ILD, echo spectrum and pulse-echo delay, and ILDs alone are only the key mechanism for horizontal sound localization.

Line 101: What is meant by “essence of appropriate pinnae motions”?

Line 102: While it is well possible to record precise 3D ear movements in bats with technology available today (Yin & Müller, Proc Natl Acad Sci U S A, 2019, 116(25):12270-12274), studying their effect on bat behavior is indeed less trivial. Nevertheless, recordings of natural pinna movements should at least be used to test whether the simulated ear movements used in the present study can actually be performed by the bats. What is the advantage of knowing which kind of ear movement would be ideal for target localization when the bats are not able to perform such movements due to physiological limitations? This drawback of the theoretical approach needs to be mentioned here also.

Line 108 and following: This paragraph is misleading. Please make clear that not ear movements but only simulations of ear movements were studied.

Line 130: It is completely unclear to me where the data on amplitude modulation is coming from. Was the pattern measured with the artificial ears in the present study? Please provide information on the body of evidence showing that pinna movements in R. ferrumequinum generate strong amplitude modulations in the echoes impinging on the bat’s ears during echolocation. Or clarify that this is only an assumption.

Line 135: Was only the silent interval between the two echoes removed or were also the FM parts of the echoes removed? What was the reason for removing the silent interval between the echoes?

Line 150: I am not sure if “attitude” is the correct word for spatial orientation of the ears.

Line 152 and following: The directionality of hearing has been measured experimentally in R. ferrumequinum by Grinnell & Schnitzler, 1977 (J Comp Physiol A 116:63-76). Although the simulated ear directionality used in the present study roughly resembles the behaviorally measured hearing directionality, I wonder if it might be possible to implement the measured hearing directionality into the model instead of the simulated one to make the model a bit less artificial?

Line 162: Why 3D? The directionality of hearing is 2D, with the dimensions azimuth and elevation.

Line 350: Which threshold was assumed for considering a specific detection error still to allow for accurate detection? What degree of detection error would prevent a bat from localizing a target?

Line 392: How was “precise direction detection” defined?

Line 395: What defines a “good motion pattern”?

Line 401: Ah, ok. A detection error of 5° was used as threshold. This should be explained in the text and not only in the figure caption. Is there any behavioral relevance for this 5° threshold?

Discussion: Here the authors constantly make conclusions about their findings in relation to the natural behavior of bats. As long as it can’t be proven that the simulated ear movements indeed resemble ear movements performed by bats, these conclusions are not valid. I suggest that the authors in general tone down the language throughout the discussion. They may consider to drop the big statements and to focus mostly on what can indeed be concluded from their findings. The discussion is very short and could be expanded. For example, the limitations of the study should be discussed in more detail. For instance, it could be mentioned that an actual bat ear is a flexible structure not a rigid plane as simulated in the model. During natural movements, the ear is likely not only rotated but also deformed and bended in itself, which might have a strong additional effect on the hearing directionality.

Line 483: This sentence includes several statements about bat behavior for which no evidence exists. 1) Why do the authors assume that the ear movements are performed by bats intentionally? It might as well be possible that the ear movements are physiologically coupled to the motor act of call production. 2) When was shown that ear movements are used by bats for target localization in 3D? To my knowledge, it has been clearly demonstrated by previous studies that ear movements in CF-FM bats are essential for target localization in elevation but not for target localization in azimuth or in the depth dimension. 3) Why do the authors assume that ear movements are energetically demanding?

Line 488: simulated motion patterns.

Line 491: 2D instead of 3D.

Line 502: What message do the authors want to convey with this sentence?

Line 505: In total 216 different simulated ear movements were tested. So, it should read “five out of 216 motion patterns”.

Line 506: No! This does not suggest that the bats select one of these five motion patterns. First, it needs to be proven that the 3D ear motion pattern of bats indeed resembles one of the five simulated patterns. And even if this would be the case, this does not show that the bats select this type of motion. They could for example due to physiological limitations only be able to perform this certain pattern of ear movement.

Line 508: Which surgical plane? Do the authors mean the midsagittal plane?

Line 513: If some support for this hypothesis could be presented, the paper would be much stronger. When watching the slow-motion video clip of Hipposideros ear movements provided with the paper by Yin & Müller (http://movie-usa.glencoesoftware.com/video/10.1073/pnas.1901120116/video-1), it seems as the ear movements consist only of a combination of yaw-axis and pitch-axis but not of roll-axis rotations.

Line 525: Well, a movement consisting of rotations along three axes, could in my eyes be considered complicated.

Figure 1: It should be mentioned in the figure caption that panel B already shows the output of the model, and which exact movement produced this specific pattern of amplitude modulation and resulting ILD pattern. Why is the modulation pattern for the forward movement different from the backward movement? Shouldn’t both patterns be congruent for each ear? When the right ear is moved forward, it is at a certain point in time aligned with the beam axis of the echo, which yields maximum amplitude. Given that the position of the bat and the target doesn’t change between the forward and the backward movement, the signal envelope should show a convex pattern also for the backward movement.

Figure 2: In panel B, the red dot with the word microphone next to it is confusing. Does the red dot represent the target in relation to the microphones? The authors may consider to choose a different color for the representation of the microphones and the target. In the figure caption it should be mentioned that the red dots indicate the position of the microphones.

Figure 4: Labeling of panels (A1 - A3, B1 – B3) is missing.

Figure 5: Labeling of panels (A1 – A3, B1 – B3) is missing.

Figure 9B: x-axis label = time [sec], y-axis label = ILD [dB SPL]

**Have the authors made all data and (if applicable) computational code underlying the findings in their manuscript fully available?**

Reviewer #1: Yes

Reviewer #2: **No: **Using the link provided with the data availability statement generates an error message.

PLOS authors have the option to publish the peer review history of their article (what does this mean?). If published, this will include your full peer review and any attached files.

Reviewer #1: No

Reviewer #2: No
---

## [Decision Letter · Decision Letter 1]

8 Jun 2022

Dear Dr. Yamada,

Thank you very much for submitting your manuscript "Theoretical investigation of active listening behavior based on the echolocation of CF-FM bats" for consideration at PLOS Computational Biology.

As with all papers reviewed by the journal, your manuscript was reviewed by members of the editorial board and by several independent reviewers. In light of the reviews (below this email), we would like to invite the resubmission of a significantly-revised version that takes into account the reviewers' comments.

We cannot make any decision about publication until we have seen the revised manuscript and your response to the reviewers' comments. Your revised manuscript is also likely to be sent to reviewers for further evaluation.

Sincerely,

Dan Stowell

Associate Editor

PLOS Computational Biology

Natalia Komarova

Deputy Editor

PLOS Computational Biology

Reviewer's Responses to Questions

**Comments to the Authors:**

Reviewer #2: 2nd review repport for PCOMPBIOL-D-21-02303 by Hiraga et al.

I thank the authors for the detailed replies to the reviewers’ comments. The revised manuscript is of significantly increased quality. However, some issues still remain.

1) Some parts of the manuscript are still hard to understand due to the incorrect usage of the English language. I highly recommend to take advantage of a professional language editing service.

2) I feel that the replies to the reviewers’ comments, which the authors have provided in the response letter, are often much more detailed (and helpful) than the changes that have been made to the manuscript itself. Many of the explanations in the response letter should also be included in the manuscript. For example, both reviewers raised questions, which the authors, as they state in the response letter, want to tackle with future experiments. This should be included in the manuscript, for example within a paragraph “Future Directions” at the end of the discussion.

3) It is still not clear to me what the assumption that ear movements produce amplitude modulations in the echoes impinging on the bats ears is based on. The authors cite an electrophysiological study demonstrating that spatial sensitivity of midbrain auditory neurons shifts in the vertical plane when the orientation of the pinna is changed. However, this study does not show that amplitude modulations are responsible for the shifts in spatial tuning of the neurons, and also does not prove that the pinna movements produce amplitude modulations in the echoes. The authors should explain in detail why they assume that for example the ear movement shown in Fig. 1B produces a maximum modulation depth of 50% of the echo amplitude. Is the maximum value of modulation depth equal for all different simulated ear movements? What is this maximum value based on? Was it measured or just assumed? Please clarify this in the manuscript.

4) Throughout the manuscript and supplementary material there are still instances where the word “attitude” is used to describe the spatial orientation of an ear. This should be rewritten.

Line-specific comments:

Line 32: The term “three-dimensional” is only used once within the abstract. Therefore, the abbreviation “(3D)” does not need to be introduced in line 32.

Line 44: … amplitude modulation in the echoes caused by ear movements.

Line 47: “allow for” instead of “accomplish”.

Line 50: “provide the conditions for ear motions to ensure accurate and robust direction detection,” can be removed entirely, since this part of the sentence is repeated in the following part of the sentence. It is sufficient to write: “In addition, we suggest that simple shaped hearing directionality and well-selected uncomplicated ear motions …”

Line 62: “control law” sounds a bit awkward. Why not “… but the precise type of ear movements that facilitate direction detection…”

Line 68-69: “The theory suggests that certain ear motions might enable highly accurate direction detection that is robust to observation errors.”

Line 81: Not all echolocators use pulses as sonar signals. Maybe “sound emissions” is a better term.

Line 95 and throughout the ms: “echo-arrival direction detection” sounds strange. Maybe use “echo source direction detection” instead?

Line 112-114: This sentence is hard to understand. Maybe rewrite to: “Obstacle avoidance experiments using bats flying in a chamber that contained multiple vertically and horizontally suspended wires showed that elevation angle detection performance is significantly decreased when bats are prevented from moving their pinnae [27].”

Line 122: Please replace “control law” by e.g. “precise nature”.

Line 124: “… the optimality in ears’ motion by physiological measurements only.”

Line 132-135: This is a very complicated sentence and hard to understand. Maybe the authors might consider to split the sentence into two shorter sentences.

Line 136-137: “The integration of such a pinnae control theory with physiological and behavioral findings might give an interpretation of bat behavior, …”

Line 139-142: “Based on these motivations, rigid ear movements for echo source detection based on ILD were mathematically investigated through a series of simulations. The simulations were conducted under the assumption that bats accomplish echo source detection by using the amplitude modulation induced by ear movements in the CF part of the echoes.”

Line 143: “Distance detection of echo sources with the FM part of the echoes is omitted from …”

Line 144-146: “In particular, simulations of different ear motions were analyzed to identify the nature of ear motions suited best for sound source detection.”

Line 180-182: This is not a very convincing argument for removing the gap between the two successive echoes. If the results are the same for simulations including the gap and simulations excluding the gap, I would prefer seeing the results with the gap present over seeing results of simulations without a gap between the two echoes. Removing the silent interval between echoes generates a very unnatural acoustic stimulus, which will never be available to a real bat. By doubling the duration of the CF part of an echo, a bat would be provided with an unnaturally long time to extract spatial information from the echo, which in theory could amplify sound source localization. To make the results more comparable to a natural situation, the input to the model should be kept as naturalistic as possible.

Line 202-207: Based on this description, I am still confused whether this was a real setup or a simulation. The abstract reads as if physical microphones have been used to measure echoes and their output has been used as input for the model. However, I assume this was not the case. Maybe this is clear to computational scientists, but for readers not so familiar with mathematical models, this paragraph should be somehow rewritten in a way that makes clear that there was no physical microphone or any sound recordings were involved. Maybe it will be helpful to write “simulated ears” instead of “directional ears”?

Line 212-213: I don’t understand this sentence.

Line 256-257: To simplify this sentence, the part after the comma can be removed: “In this section, we prepare a general mathematical framework for describing and evaluating the active sensing process.”.

Line 381-382: In my opinion, it is not correct to state that bats do not perform complex ear movements. I as a human being who is not able to move its ears at all, consider the ear movements performed by bats as highly complex.

Line 549-551: “In this study, simulations of rigid ear movements for direction detection with ILD were mathematically investigated by assuming that ear movements create useful amplitude modulations on the echoes perceived by bats.”

Line 552: “… three-axis rotations can be useful for accurate echo direction detection.”

Line 559: remove the word “measurement”

Figure 2A: What is indicated by the two grey dots located at the circle in the horizontal plane?

**Have the authors made all data and (if applicable) computational code underlying the findings in their manuscript fully available?**

Reviewer #2: Yes

PLOS authors have the option to publish the peer review history of their article (what does this mean?). If published, this will include your full peer review and any attached files.

Reviewer #2: No
---

## [Decision Letter · Decision Letter 2]

26 Sep 2022

Dear Dr. Yamada,

We are pleased to inform you that your manuscript 'Theoretical investigation of active listening behavior based on the echolocation of CF-FM bats' has been provisionally accepted for publication in PLOS Computational Biology.

Best regards,

Dan Stowell

Academic Editor

PLOS Computational Biology

Natalia Komarova

Section Editor

PLOS Computational Biology

Reviewer's Responses to Questions

**Comments to the Authors:**

Reviewer #2: Review repport for PCOMPBIOL-D-21-02303R2 by Hiraga et al.

I thank the authors for addressing all my previous comments and support publication of their paper in PLOS Computational Biology.

**Have the authors made all data and (if applicable) computational code underlying the findings in their manuscript fully available?**

Reviewer #2: Yes

PLOS authors have the option to publish the peer review history of their article (what does this mean?). If published, this will include your full peer review and any attached files.

Reviewer #2: No

---

## [Editor Report · Acceptance letter]

30 Sep 2022

PCOMPBIOL-D-21-02303R2 

Theoretical investigation of active listening behavior based on the echolocation of CF-FM bats

Dear Dr Yamada,

I am pleased to inform you that your manuscript has been formally accepted for publication in PLOS Computational Biology. Your manuscript is now with our production department and you will be notified of the publication date in due course.

With kind regards,

Anita Estes
